# Plasmonic Nanomaterials for Micro- and Nanoplastics Detection

Serena Schiavi [1], Miriam Parmigiani [1], Pietro Galinetto [2], Benedetta Albini [2], Angelo Taglietti [1] and Giacomo Dacarro [1,3,*]

1 Dipartimento di Chimica, Università degli Studi di Pavia, Via Taramelli 12, I-27100 Pavia, Italy; serena.schiavi01@universitadipavia.it (S.S.); miriam.parmigiani01@universitadipavia.it (M.P.); angelo.taglietti@unipv.it (A.T.)
2 Dipartimento di Fisica, Università degli Studi di Pavia, Via Bassi 6, I-27100 Pavia, Italy; pietro.galinetto@unipv.it (P.G.); benedetta.albini@unipv.it (B.A.)
3 Centre for Health Technologies (CHT), Università degli Studi di Pavia, I-27100 Pavia, Italy
* Correspondence: giacomo.dacarro@unipv.it; Tel.: +39-382987337

**Abstract:** Detecting and quantifying micro- and nanoplastics (MNPs) in the environment is a crucial task that needs to be addressed as soon as possible by the scientific community. Many analytical techniques have been proposed, but a common agreement on analytical protocols and regulations still has to be reached. Nanomaterial-based techniques have shown promising results in this field. In this review, we focus on the recent results published on the use of plasmonic noble metal materials for the detection of MNPs. Plasmonic materials can be exploited in different ways due to their peculiar optical end electronic properties. Surface plasmon resonance, plasmon enhanced fluorescence, UV–Vis spectroscopy, and surface enhanced Raman scattering (SERS) will be considered in this review, examining the advantages and drawbacks of each approach.

**Keywords:** microplastics; nanoplastics; SERS; Raman; plasmonics; noble metal nanoparticles





## 1. Introduction

The global plastics production was estimated to be over 350 million metric tons per year in 2021, and is still growing [1]. In parallel, the market value of these materials is constantly increasing. Most of the industrially produced plastics are discarded into the environment, and for the majority of this waste, the destiny is still uncertain: 8% is discarded in landfills, 9% is recycled, and 12% is used for energy recovery. The rest, more than two thirds, is released and accumulated in the environment, with rivers and oceans being the main destination [1]. Released plastics can have different sizes, ranging from bulk objects down to the nanometric scale. Objects in the micrometric and nanometric size range are commonly classified as micro- and nanoplastics, respectively (MNPs). The upper and lower limits of these categories are not universally recognized [2]. Regarding microplastics, the commonly accepted upper limit is 5 mm, even if some authors set 1 mm as the superior limit of this category. The European Chemical Agency (ECHA) defines microplastics as particles below 5 mm in size, which also includes fiber-like particles that are between 5 and 15 mm in length [3]. The same limit is set by the National Oceanic and Atmospheric Administration in the U.S. [4]. The definition of nanoplastics is even more uncertain: according to the European Food Safety Authority (EFSA) panel on contaminants in the food chain (CONTAM), nanoplastics range from 1 to 100 nm in size [5], in compliance with the ISO definition of nanomaterials, which states that a nanomaterial is a material with any external dimension in the nanoscale or having an internal structure or surface structure in the nanoscale. The nanoscale is defined as ranging from approximately 1 to 100 nm (0.001–0.1 μm), but some papers have set the upper size limit for nanoplastics to 1 μm [6]. On the other hand, in order to establish a regulation, it may be necessary to raise the lower size limit of nanoplastics to 5 nm, since no techniques are currently available to detect smaller particles.

The origins of MNPs dispersed in the environment are different: the so called primary MNPs are intentionally produced polymeric particles under 5 mm in size, like in the case of personal care products or precursors of plastic products. Secondary MNPs, on the other hand, are generated by the fragmentation and erosion of bigger plastic wastes. Thus far, EU regulations have mainly targeted the use of primary MNPs in order to reduce their production and commercialization: in January 2019, the ECHA proposed restricting intentionally added microplastics [7]. In order to reduce the impact of petroleum-derived plastics to the environment, the use of bioplastics is continuously increasing, since bioplastics can be used as substitutes for traditional plastics in several fields [8,9], or even as materials for MNP removal [10].

Only recently (2020) has the control of microplastics in drinking water become a priority for the European Union (EU directive 2020/2184) "on the quality of water intended for human consumption". This directive demands the adoption of a methodology to measure microplastics by January 2024, since no standard procedure has been set thus far.

While MNPs were initially thought to accumulate mainly in seawater, in the last years, evidence has been reported for their presence in almost every area of the environment: fresh water [11,12], tap water [13,14], soil [15], air [16], plants [1,17], ice [18], etc.

Another crucial aspect that needs to be addressed in the next few years is the lack of data on the exposure and risk for human health [19]. At present, a complete human health risk assessment (HRA) is not possible due to the lack of data on human exposure to MNPs, but a recent study showed the presence of microplastics in human blood [20], increasing the concerns on the persistence of this pollutant in the body. Recent data have also reported on the presence of microplastics in human placenta [21]. Data are also lacking on the presence of MNPs as contaminants in food [5], and the EFSA has expressed the need for the standardization of analytical techniques for microplastics analysis and for the establishment of new techniques for nanoplastics.

So far, several instrumental techniques have emerged for the analysis of MNPs, with many dedicated papers and reviews. We provide a brief overview of these techniques in the next section, before focusing on the main topic of this review: the use of plasmonic nanomaterials for MNP sensing and analysis. For a more in depth discussion on the other techniques, the reader can refer to the literature and the references provided.

## 2. Overview of MNPs Detection Techniques

The detection and quantification of MNPs are recent and emerging research topics, growing at fast pace as the problem of detecting and removing these harmful materials is constantly rising in importance, raising serious concerns.

Analytical techniques that can be used for MNPs detection can be roughly divided into optical and spectroscopic techniques (mainly based on FTIR and Raman), fluorescence microscopy, mass spectrometry techniques (GC-MS, ICP-MS, MALDI-TOF-MS), and microscopy techniques (electronic and optical). In Table 1, we summarize the main techniques reported in the literature, providing an indication of the size range of particles that can be identified, of the possibility of conducting qualitative identification (i.e., which type of plastic is detected), and, when available, the limit of detection/quantification (LOD/LOQ). The list is not meant to be exhaustive, as many complete reviews have already been dedicated to these topics [22–25]. For each technique, a representative paper is cited for reference, but also in this case, a complete review of the available literature was out of the scope of this paper.

**Table 1.** Summary of the main analytical techniques currently used for MNPs detection and quantification.

| Technique | | Size Range | Composition | Concentration (LOD) | Limitations |
|---|---|---|---|---|---|
| Spectroscopy [26] | FTIR | 50–500 µm | YES | \\\\\\ | Not suitable for nanoplastic identification |
| | Raman | >100 nm | YES | \\\\\\ | |
| Fluorescence microscopy | NILE RED [27] | 20 µm–1 mm | NO | \\\\\\ | Not suitable for less hydrophobic polymers |
| | PDI [28] | \\\\\\ | Selective for PVC | \\\\\\ | |
| | Fluorogenic polymer [29] | 0.1–100 µm | YES | \\\\\\ | |
| MS | Py-GC/MS [30] | \\\\\\ | YES | LOD: 4 mg/L | Complex sample pre-treatment |
| | MALDI-TOF MS [31] | \\\\\\ | YES | LOD: 25 mg/L | Thermal pre-treatment |
| | Single particle ICP-MS [32] | 135 nm–1 µm | YES | $8.4 \times 10^5$ NP/L (LOQ) | NPs need to be carboxylated for their identification |
| Microscopy | SEM [33] | >100 nm | NO | \\\\\\ | No information available about composition. Usually coupled with other techniques. |
| | TEM [33] | >100 nm | NO | \\\\\\ | |
| | optical | >0.5 µm | NO | \\\\\\ | Not suitable for nanoplastic identification. Not possible to confirm the chemical composition |

## 3. Plasmonic Nanomaterials for MNPs Detection

Metal nanoparticles have a wide application in modern sensing and analytical and bioanalytical techniques, mainly due to their peculiar optical properties. In this section, we describe all of the analytical approaches relying on localized surface plasmon resonance (LSPR).

We recall that localized plasmonic resonance is the result of the confinement of a sur-face plasmon in a nanoparticle of a size comparable to or smaller than the wavelength of light used to excite the plasmon [34]. When a small spherical metallic nanoparticle is irradiated by light, the oscillating electric field causes the conduction electrons to collectively oscillate. This collective oscillation is called a localized surface plasmon. This physical entity can be used to exploit different analytical approaches for sensing. They can be generally divided into colorimetric methods (i.e., relying only on color change), SPR reflectance methods (based on the use of a SPR sensor chip), and plasmon enhanced fluorescence methods (based on the enhancement of the emission properties of the material).

### 3.1. Colorimetric Sensors

Plasmonic metal nanoparticles are commonly employed as colorimetric sensors for a wide variety of analytes and applications: food safety [35], environmental pollutants [36], cancer diagnostics [37], and the sensing of pathogens [38]. Colorimetric methods are simple and low-cost, and the response of the sensor can often be evaluated by the naked eye, without the need for complex instrumentation. Metal nanoparticle-based colorimetric sensors usually exploit an aggregation of the colloid, leading to a dramatic change in the LSPR maximum absorption wavelength [39].

To our knowledge, only one paper in literature has reported the use of gold nanoparticles for the colorimetric sensing of polystyrene (PS) nanoplastics [40]. The method is based on a partial dissolution of PS in acetone. PS chains inhibit the acetone-induced aggregation of AuNPs, leading to different colors of the dispersion in the presence and absence of nanoplastics, as shown in Figure 1. In other words, non-aggregated AuNPs maintain their typical purple-red color, signaling the presence of plastics. Aggregated AuNPs, on the other hand, shift to a blue color, diagnostic of particle aggregation and of a consequent change in the maximum absorption wavelength of the plasmon. This method is promising and provides easy to read results, but it has some major limitations, also acknowledged

by the authors, which need to be overcome in order to use it in a real application. A long time (one day) is needed for sample equilibration after the addition of acetone, no data are available yet on possible interferents, and a more quantitative interpretation of the data could be added by using spectrophotometric data in addition to a naked eye evaluation.

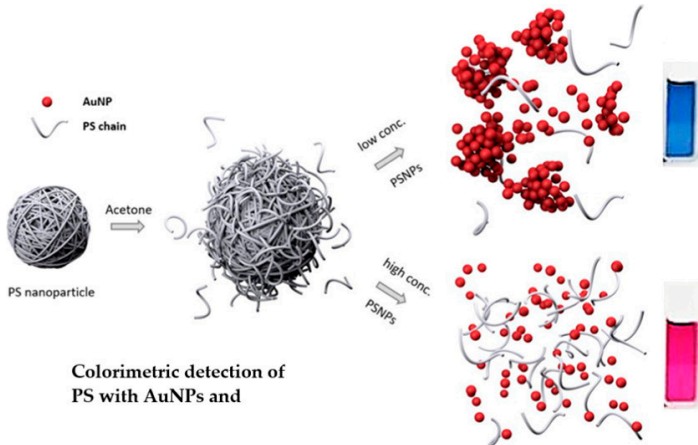

**Figure 1.** Colorimetric detection of MNPs. PS nanoparticles in acetone can be detected via the PS induced aggregation of AuNPs. Non-aggregated AuNPs show the typical deep ref colour, while aggregated colloids turn blue, as shown in the pictures in figure. Reprinted with permission from ref. [40]. Copyright Elsevier 2022.

### 3.2. Surface Plasmon Resonance

Surface plasmon resonance (SPR) sensing is a powerful analytical technique used to detect and analyze molecular interactions in real-time [41]. It is a label-free, non-destructive technique that can be used to study a wide range of molecular, ionic, and biomolecular interactions including protein interactions [42], protein–DNA interactions [43], and antibody–antigen [44] binding. SPR sensing relies on the phenomenon of surface plasmon resonance, which occurs when a thin layer of metal (typically gold or silver) or a layer of nanoparticles is grafted on a surface and excited with light. This excitation causes a collective oscillation of free electrons, known as SPR, at the metal surface. When biomolecules are immobilized on the metal surface, any changes in their mass, conformation, or refractive index can alter the SPR, which can be detected as a change in the reflected light intensity or in the maximum absorption wavelength. SPR sensing has become an important tool in drug discovery, diagnostics, and biotechnology, and its applications continue to expand as new technologies and detection methods are developed.

Although SPR sensing is most commonly associated with biosensing applications, it has also found applications in other areas. One notable example is the use of SPR in the field of materials science. SPR can be used to study the properties of thin films and coatings [45]. The study of the interactions of SPR with gases [46] and liquids has led to the development of SPR-based sensors for gas detection, environmental monitoring, and food safety [47]. SPR sensors based on gold nanostructures can be easily integrated in optic fibers, useful for the creation of compact and easy to use devices for bio-chemical detection and pollutant analysis [48,49].

In 2016, Safina et al. [50] proposed a method based on a commercial SPR setup for biosensing. A commercial SPR chip, made of a 50-nm thin film of gold, was used as the sensing chip, operating at 670 nm. The method was tested on 100 nm, 300 nm, and 460 nm nominal diameter polystyrene. The method was based on the measurement of the diameter and particle number concentration of the PS nanoparticles by fitting the measured effective refractive index of polystyrene colloids (Figure 2). Data obtained from the SPR measurements were compared to the SEM size analysis, and concentrations were compared to the concentration of the commercial standards.

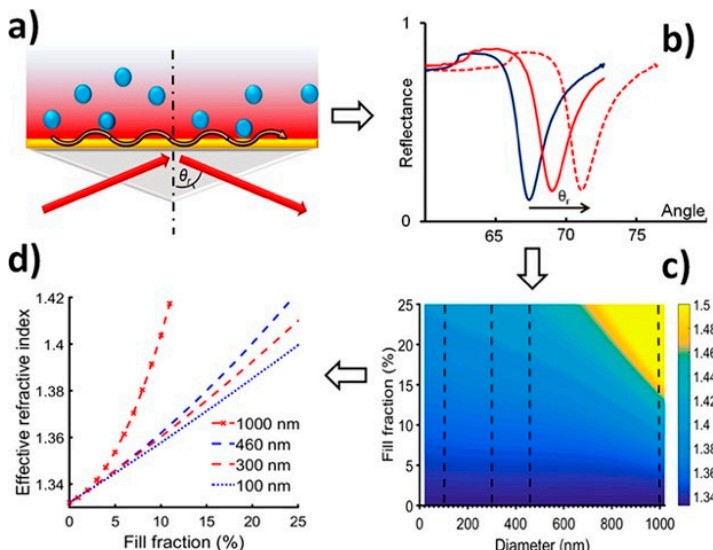

**Figure 2.** Schematic representation of the surface plasmons generated in a Au layer (**a**) and reflectance plot as a function of the angle of incidence at different dilution factors: 0.1, blue; 0.5, solid red; and 1, dashed red (**b**). From the calculated refractive indices as a function of particle diameter (dpart) and fill fraction (**c,d**), it is possible to simultaneously determine the diameter and number concentration of particles of different sizes. The dashed vertical lines indicate sizes for which the $n_{eff}$ is calculated as a function of f in (**d**). Reprinted with permission from ref. [50]. Copyright 2016 American Chemical Society.

Fitting of the effective refractive index was conducted using the coherent scattering theory (CST), and data obtained for the three standard samples were coherent with what were obtained from the validation methods. For the 100 nm sample, the measured refractive index values were higher than the prediction carried out with CST, probably due to the adsorption of the particles on the Au surface. Some problems also emerged in the size measurement of the 300 nm particles, with an SPR measured diameter lower than the diameter obtained by means of SEM microscopy. The method is thus promising, but still needs some improvement and a better understanding of the data. The feasibility of a simultaneous measurement of the diameter and concentration has still to be demonstrated, but the method shows good results in this direction.

With a similar approach, Dong et al. [51] prepared an SPR chip functionalized with estrogen receptors (ER) for the real-time detection of microplastics. With chromatography experiments, the authors demonstrated the possibility of distinguishing different microplastics (polystyrene—PS, polyvinyl chloride—PVC, polyethylene—PE with average size of 20 μm) based on their surface charge. Affinity and dissociation constants were calculated for the different microplastic types: a difference was observed for PS, PVC, and PE affinity, but it is not clear from the paper whether the method would be suitable for microplastic differentiation in water.

Oligo-peptides have also shown good affinities for MNP, being able to selectively bind specific plastics like PS, PE, or PP [52,53]. Lee and coworkers studied a sensor based on a commercial SPR chip and a UV–Vis spectrophotometer reading. A self-assembled monolayer (SAM) of 40 nm AuNPs was used on the detecting chip (using a commercial sensor chip compatible with UV–Vis spectrophotometers) and two different sensing approaches were tested: direct sensing on the chip and sandwich sensing with 5 nm AuNPs to improve the signal. A thiol-functionalized oligo peptide was grafted on gold and used as the sensing moiety. PS nanoplastics at different concentrations were tested, measuring the variation in the LSPR band intensity. A good linear response of Abs vs. PS concentration was obtained both with the chip alone and the sandwich approach, with the second yielding a higher signal. The system was also tested in flow with a recirculation module coupled to a conventional UV–Vis spectrophotometer.

### 3.3. Plasmon Enhanced Fluorescence

Plasmon-enhanced fluorescence (PEF) is a well-established technique that grants huge enhancement factors and the possibility to detect analytes even at the single molecule level. The effect of PEF was observed in the early 1980s in concurrence with the observation of the SERS effect [54]. This effect greatly enhances the emission of organic molecules in the proximity of plasmonic nanostructures, allowing for the detection of weakly emitting species that are not detectable otherwise [55]. PEF has been applied to single-molecule detection [56], the detection of DNA [57], and cell imaging [58].

Recently, an application has also been envisaged for the visualization of microplastics and fibers [59]. The authors used a gold nanopillar substrate previously engineered as a SERS substrate [60]. The method was tested with low-density polyethylene (LDPE), poly(butylene adipate-co-terephthalate) (PBAT), and epoxy resins: a drop of a microplastic-containing water sample was dried on the substrate and analyzed by means of fluorescence microscopy. PEF allows one to visualize the PBAT and LDPE particles and fibers without the need of a dye. The larger fibers were also detected with optical microscopy, but the use of PEF allowed for the detection of much smaller particles, otherwise invisible to optical microscopy.

The LOD and LOQ for a microplastic sample in milliQ water were calculated to be 0.35 and 1.2 femtograms, respectively (samples were generated by exposing a plastic film to MilliQ water at 23 °C for 8 weeks, according to a previous published procedure [61]). An enhancement factor of 68 was determined by comparing the signal-to-noise ratio of the fluorescence of microplastics on the nanopillar substrate, related to the S/N ratio of microplastics on bare glass. The method also proved to be reliable on seawater samples, allowing for the detection of microplastics fibers, even when surrounded by salt crystals generated after drying. This paper could be an interesting proof-of-principle for further studies on the use of PEF for micro- and nanoplastics detection.

### 3.4. Advantages and Disadvantages of Plasmonic Nanomaterials

The employment of plasmonic nanomaterials for MNPs detection is a completely novel topic, accounting for a limited number of papers (cited in the previous paragraphs) published in the last 2–3 years. All of these papers are proof-of-principle studies, opening up interesting perspectives for a wider use of these materials.

Colorimetric methods can provide an easy to use technique that can be applied in the field without the use of expensive instrumentation. On the other hand, these methods are intrinsically unable to provide reliable quantitative information and will hardly be able to compete with an instrumental method. Colorimetric assays, however, are useful in many fields for fast and qualitative screening, and could also find employment in the analysis of MNPs.

SPR sensing can rely on a solid base of instruments and chips already available in the market for other purposes, mainly for biological assays. This could offer a good starting point for the development of specific methods for MNPs detection and quantification. To date, however, only a couple of proof-of-principle papers have been published on this topic.

PEF also looks promising, exploiting the good S/N ratio of enhanced fluorescence. The employment of PEF, however, has only been reported in one paper in the literature to our knowledge.

Some of the aforementioned papers showed promising preliminary data on the discrimination of plastics with different sizes and different composition. The methods, however, still need to be tested on complex and real samples. The main issue that needs to be addressed is the performance of this methods on real samples and matrices.

## 4. SERS-Based MNPs Detection

As mentioned earlier in the introduction, the detection of plastic particles in the nanometric and micrometric range is still challenging for most techniques, but it is also an urgent issue that needs to be resolved.

SERS is a very sensitive technique with a great potential for the detection of trace analytes [62,63]. In fact, the SERS technique enhances the weak Raman signals of the

probe molecules thanks to the hot spot generated from a nano-metallic substrate at close distance [64]. At the same time, the SERS sensor can also identify chemical and biological species of different natures thanks to their diagnostic Raman fingerprints [65,66]. SERS can thus provide qualitative and quantitative information that are useful for detecting the presence of MNPs in real samples, and even for discriminating between the different polymeric materials, by analyzing their different Raman signatures in the spectrum. The typical Raman active peaks mode for common plastic materials (i.e., polystyrene—PS, polymethyl-methacrylate—PMMA, polyethylene terephthalate—PET, polycarbonate—PC, polypropylene—PP, polyvinyl chloride—PVC) are reported in Table 2, together with their assignment. The most relevant and diagnostic peaks for each material are highlighted in bold in the table.

**Table 2.** Typical Raman peaks of the different microplastics. PS: polystyrene. PMMA: polymethyl-methacrylate. PET: polyethylene terephthalate. PC: polycarbonate. PP: polypropylene. PVC: polyvinyl chloride. The most relevant modes are reported in bold.

| Polymer | Wavenumber (cm$^{-1}$) | Assignment |
|---|---|---|
| PS [67–90] | 616 | In-plane aromatic ring deformation |
| | 795 | –C–H out-of-plane deformation |
| | **1003** | **Aromatic ring C–C stretching mode** |
| | **1030** | **C–H in-plane deformation** |
| | 1150 | C–C stretching mode |
| | 1205 | $C_6H_5$–C vibration |
| | 1310 | $CH_2$ in-phase twist |
| | 1352 | CH deformation |
| | 1410 | Ring stretching |
| | 1455 | $CH_2$ bending |
| | 1527 | Ring stretching |
| | 1583 | C=C stretching |
| | 1600 | Ring skeleton stretching |
| | 2915 | Asymmetric $CH_2$ stretching |
| | 3060 | Aromatic CH stretching |
| PMMA [67,68,74,78,81,88, 90–93] | 602 | C–COO stretching, C–C–O symmetric stretching |
| | **817** | **C–O–C symmetric stretching** |
| | 878 | $CH_2$ stretching |
| | 925 | $CH_2$ stretching |
| | 1000 | O–$CH_3$ rock |
| | 1081 | Stretching C–C skeletal mode |
| | 1264 | C–O stretching, C–COO stretching |
| | 1460 | C–H asymmetric bending of $\alpha$–$CH_3$ and C–H asymmetric bending of O–$CH_3$ |
| | 1648 | Combination band involving C=C stretching and C–COO stretching |
| | **1736** | **C=O stretching of C–COO** |
| | 2848 | Combination band involving O–$CH_3$ |
| | **2957** | **C–H stretching of O–$CH_3$, C–H asymmetric stretching of$\alpha$–$CH_3$ and asymmetric stretching of $CH_2$** |
| | 3002 | C–H asymmetric stretching of O–$CH_3$ and C–H asymmetric stretching of $\alpha$–$CH_3$ |
| | 3454 | Overtone of 1736 cm$^{-1}$ |
| PET [74,84,94,95] | 633 | C=C symmetrical bending vibration of the benzene ring |
| | **854** | **C=C stretching vibration between the benzene ring and the carboxyl group** |
| | 1116 | Ester C(O)–O and ethylene glycol C-C bonds |
| | 1178 | Ring in-plane C–H bond and C–C stretching |
| | 1288 | C(O)–O stretching |
| | 1308 | Ring C–H in-plane bending |
| | 1414 | C–C–H bending and O–C–H bending |
| | 1452 | $CH_2$ bending and O–C–H bending |
| | **1613** | **C=C stretching vibration of the carboxyl group** |
| | 1724 | C=O stretching vibration |
| | 2895 | C–H bond of methylene sequences |
| | 2960 | Methylene groups nearby oxygen atoms |

**Table 2.** *Cont.*

| Polymer | Wavenumber (cm$^{-1}$) | Assignment |
|---|---|---|
| PE [74,77,96–101] | 1060 | Symmetric C–C stretching mode |
| | 1080 | C–C stretching |
| | 1127 | Asymmetric C–C stretching mode |
| | 1170 | CH$_2$ rocking |
| | **1291** | **CH$_2$ twisting mode** |
| | 1408 | CH$_2$ bending and CH$_2$ wagging |
| | 1429 | CH$_2$ symmetric deformation |
| | **1450** | **CH$_2$ scissoring** |
| | 2838 | Symmetric CH$_2$ stretching |
| | 2882 | Asymmetric CH$_2$ stretching |
| PC [74,102–108] | 405 | O–C–O bending |
| | 578 | In-plane ring deformation |
| | 630 | In-plane ring deformation |
| | 705 | Out-of-plane ring deformation |
| | 734 | C-ring stretching |
| | 830 | Out-of-plane ring skeleton stretching |
| | **888** | **[O–(C=O)–O] stretching, C–CH$_3$ stretching, CH$_3$ rocking, and out-of-plane ring skeleton stretching** |
| | 920 | C–CH$_3$ stretching and –CH$_3$ rocking |
| | 944 | C–H in-plane bending |
| | 1005 | Ring skeleton stretching |
| | **1109** | **–C–H ring in-plane bending** |
| | 1121 | C-O-C stretching |
| | 1175 | C–O–C stretching and –C–H ring in-plane bending |
| | 1238 | C–O–C asymmetric stretching, CO deformation, and C-ring stretching |
| | 1297 | C–O–C stretching |
| | 1320 | C–O–C stretching |
| | 1396 | CH$_3$ deformation |
| | 1448 | CH$_3$ symmetric deformation |
| | 1467 | CH$_3$ asymmetric deformation |
| | 1592 | Ring skeleton stretching |
| | 1606 | Ring skeleton stretching |
| | 1777 | C=O stretching |
| | 2070 | CH$_3$ stretching |
| | 2106 | CH$_2$ symmetric stretching |
| | 2200 | CH$_2$ asymmetric stretching |
| | 2218 | CH$_3$ asymmetric stretching |
| | 2950 | Stretching of different C–H$_n$ groups |
| PP [74,109] | 252 | CH$_2$ wagging and CH bending |
| | 321 | CH$_2$ wagging |
| | 398 | CH$_2$ wagging and CH bending |
| | 458 | CH$_2$ wagging |
| | 530 | CH$_2$ wagging, stretching C–CH$_3$ and CH$_2$ rocking |
| | **809** | **CH$_2$ rocking, C–C stretching, and C–CH$_3$ stretching** |
| | 841 | CH$_2$ rocking, C–C stretching, C–CH$_3$ stretching, and CH$_3$ rocking |
| | 900 | CH$_3$ rocking, CH$_2$ rocking, and CH bending |
| | 941 | CH$_3$ rocking and C–C stretching |
| | 973 | CH$_3$ rocking and C–C stretching |
| | 998 | CH$_3$ rocking, CH bending, and CH$_2$ wagging |
| | 1040 | C–CH$_3$ stretching, C–C stretching, and CH bending |
| | **1152** | **C–C stretching, C–CH$_3$ stretching, CH bending, and CH$_3$ rocking** |
| | 1219 | CH$_2$ twisting, CH bending, and C-C stretching |
| | 1330 | CH bending and CH$_2$ twisting |
| | 1360 | CH$_3$ symmetric bending and CH bending |
| | **1458** | **CH$_3$ asymmetric bending and CH$_2$ bending** |
| | 2840 | CH$_2$ symmetric stretching |
| | 2883 | CH$_3$ symmetric stretching |
| | 2905 | CH stretching |
| | 2920 | CH$_2$ asymmetric stretching |
| | 2952 | CH$_3$ asymmetric stretching |

**Table 2.** *Cont.*

| Polymer | Wavenumber (cm$^{-1}$) | Assignment |
|---|---|---|
| PVC [74,110–113] | 361 | C–Cl in-plane bending (in trans HClC=CHCl) |
| | **625** | **Crystalline C-Cl (in Cl$_2$C=CHCl) stretching** |
| | **686** | **C–Cl symmetric stretching (in H$_2$C=CHCl)** |
| | 961 | C–Cl asymmetric stretching (in H$_2$C=CHCl) |
| | 1100 | C–O stretching |
| | 1179 | CH$_2$ twisting |
| | 1255 | –C–H rocking |
| | 1325 | CH$_2$ deformation and –C–H bending |
| | **1420** | **CH$_2$ symmetric deformation** |
| | 1724 | Ester CO stretching |
| | 2819 | –C–H stretching vibration |
| | 2851 | CH$_2$ symmetric stretching |
| | 2914 | CH$_2$ asymmetric stretching |
| | 2940 | CH$_2$ asymmetric stretching |
| | 2975 | –C–H stretching |

### 4.1. The SERS Effect, Enhancement Factors, and Qualitative Information

The SERS effect is the result of the interplay between the physical and chemical surface processes occurring in the molecule–nanostructure system [114–116]. However, the predominance of one or more of these mechanisms is hardly linked to the type of nanostructures used as SERS substrates.

As highlighted by the overview of recent SERS works on plastics detection reported in Table 3, the most widely used substrates are composed of nanostructured metals. Thus, the description on the mechanisms at the bases of the SERS effect will only be present for metallic systems, since it was beyond the aim of this review to also report a detailed illustration of all the SERS theories explaining the effect on dielectric and/or hybrid systems. However, it is worth mentioning this type of substrate since they are increasingly occupying a prominent place in the SERS landscape [117].

Let us consider the simplest and illustrative case of a single molecule excited by monochromatic incident light. The inelastic component of the radiated scattered light by the molecule itself corresponds to the Raman scattering. The source of this phenomenon is the generation of an induced oscillating dipole in the molecule due to the displacement of the electron cloud with respect to the positive charges, in response to an applied external electric field, that from a classical point of view is given by: $\overline{p} = \overline{\overline{\alpha}} \cdot \overline{E}_0$, where $\overline{\overline{\alpha}}$ is the polarizability, a second-rank tensor, and $\overline{E}_0$ is the incident electric field vector.

If we now consider a metallic nanosphere in the proximity of our molecule, the equation for the induced electric dipole still holds, and it can be re-written as: $\overline{p}^* = \overline{\overline{\alpha}}^* \cdot \overline{E}_{loc}$.

This simple equation encloses the two main mechanisms at the basis of the enhancement. Indeed, $\overline{E}_{loc}$ encloses the so-called electromagnetic enhancement (EE), occurring when the incident frequency matches the LSPRs. The term, $\overline{\overline{\alpha}}^*$, is the modified polarizability due to the interaction of the molecule with the metal nanosphere. According to [118], all modifications to the polarizability tensor due to the adsorption of the molecule on the metal surface correspond to the chemical enhancement (CE).

A key parameter that allows us, in some way, to quantify the enhancement of the Raman signal, is the so-called enhancement factor (EF). EF is theoretically defined as the ratio between SERS and the Raman (RS) intensity of the same investigated analyte. Neglecting the frequency shifts between incident and Raman light (this holds in particular for modes at low Raman shifts), the theoretical expression for EF is given by the well-known approximation to the fourth power of the local field:

$$\text{EF} = \frac{I_{\text{SERS}}}{I_{\text{RS}}} \approx \left| \frac{\alpha_{\text{SERS}}}{\alpha_{\text{RS}}} \right| \left| \frac{E_{\text{loc}}}{E_0} \right|^4 \tag{1}$$

However, a currently still open issue is the lack of a universal operative formula for the practical estimation of the EF. The variety of different approaches to the EF calculation is also highlighted by the literature reported herein. In the following, we illustrate the most commonly used EF formulations in the papers cited in this review (Equations (2)–(5)), although an exhaustive treatment on the different common operative formula can be found in [119].

$$EF = \frac{I_{SERS}/N_{SERS}}{I_{NRS}/N_{NRS}} \tag{2}$$

$$EF = \frac{I_{SERS}/C_{SERS}}{I_{NRS}/C_{NRS}} \tag{3}$$

$$EF \approx \int_v \frac{|E_{loc}(\omega_0, r)|^4}{|E_0(\omega_0, r)|^4} dv * \tag{4}$$

$$EF = \frac{I_{SERS}}{I_{NRS}} \tag{5}$$

where $I_{SERS}$ is the intensity of the selected SERS peak; $I_{NRS}$ is the intensity of the selected normal Raman peak; $N_{SERS}$ is the number of particles contributing to the intensity of SERS signal; $N_{NRS}$ is the number of particles contributing to the intensity of the normal Raman signal; $C_{SERS}$ is the sample concentration in SERS measurements; $C_{NRS}$ is the sample concentration in Raman measurements; $E_{loc}$ is the E-field at the scatterer (e.g., E-field in the presence of metal nanoparticles); $E_0$ is the incident field at the position of molecules of interest (E-field in the absence of metal nanoparticles).

* It must be stressed that this equation is only applied in theoretical studies.

In our research works [120,121], we proposed another declination of the general definition of Equation (1), with $N_{SERS}$ and $N_{RS}$ defined as the product between the scattering volume and molecular concentrations in the two cases ($c_{SERS}$ and $c_{RS}$), respectively. In particular, since we used the same experimental configuration for both the Raman and SERS experiments, namely, a wet drop between two blank glass slides or, for SERS, one blank glass slide and the SERS substrate, the investigated area was the same and the only operative parameter that changed was the height of the scattering volume. Thus, our operative formula can be written as

$$EF = \frac{I_{SERS}}{I_{RS}} \frac{c_{RS}H_{RS}}{c_{SERS}H_{SERS}} \tag{6}$$

where $H_{SERS}$ is about 10 nm, since for a greater distance, the SERS enhancement drops [120,122], while $H_{RS}$ is equal to the interspace between the two blank glass slide, namely, the height of the spread drop, since in the Raman effect, all of the molecules contribute to the recorded signal.

### 4.2. Detection of MNP Using SERS-Based Techniques

Even though the application of SERS for micro and nanoplastics detection is still in its infancy, several works have been published on this topic, with increasing numbers over the last years. Among the different techniques based on plasmonic materials that we have cited so far, SERS-based approaches are receiving the greatest interest in the scientific literature.

In Table 3, we tried to summarize the different examples of SERS-based materials for MNPs sensing and detection that can be found in the literature. The main characteristics and the relevant experimental parameters of each approach (i.e., the operating laser wavelength, the type of detected plastics, the limits of detection where available, etc.) are included in the table, while a more detailed discussion is reported in the following paragraphs.

**Table 3.** SERS-based approaches to MNPs detection and sensing.

| Substrate | Plastics | EF | $\lambda_{exc}$ (nm) | LOD (µg/mL) | Real Sample LOD |
|---|---|---|---|---|---|
| Colloidal | | | | | |
| AgNPs [123] | PS 100, 500 nm PE 10 µm PP 10 µm | PS 100 nm: EF = $5 \times 10^2$ PS 500 nm: EF = $4 \times 10^4$ in seawater: PS 100 nm: EF = $7 \times 10^2$ PS 500 nm: EF = $1.1 \times 10^4$ (Equation (3)) | 785 nm | PS 100, 500 nm: 40 µg/mL | PS 100 and 500 nm in seawater: 40 µg/mL |
| AgNPs [68] | PS 1 µm, 50 nm | -- | 785 nm | PS 1 µm, 50 nm: 5 µg/mL | 1 µm, 50 nm: 5 µg/mL in river water |
| AuNPs [71] | PS 20(+), 20(−), 200(+), 200(−) nm | PS 20(+) nm: EF = 3050 (Equation (3)) | 785 nm | PS 20(+) nm: 10 µg/mL PS 200(+) nm: 1 µg/mL | PS 20(+) nm: 100 µg/mL in NaCl 150–600 mM solution and seawater |
| Au nanourchins [124] | PS 600 nm | -- | 785 nm | -- | -- |
| AuNPs [82] | PS 100 nm | -- | 532 nm | 200 µg/mL | -- |
| AgNPs [76] | PS 50 to 2 µm | PS 50 nm: EF = $1 \times 10^4$ PS 100 nm: EF = $2.30 \times 10^4$ PS 200 nm: EF = $5.50 \times 10^3$ PS 500 nm: EF = $6 \times 10^3$ (Equation (3)) | 785 nm | PS 50 nm: 12.5 µg/mL PS 100 nm: 6.25 µg/mL PS 200 nm: 25 µg/mL PS 500 nm: 25 µg/mL PS 1 µm: 12.5–25 µg/mL | PS 50 nm: 50 µg/mL PS 100 nm: 200 µg/mL PS 200 nm: 200 µg/mL PS 500 nm: 100 µg/mL In real water |
| Elliptical AuNPs [77] | PS 350 nm, 1 µm PE 1–4 µm | -- | 632 nm | PS 350 nm: 6.25 µg/mL | -- |
| Au nanorods [125] | PS 100 nm-1 µm | R6G: EF = $5.4 \times 10^5$ (Equation (2)) | 785 nm | PS 100 nm: 1 µg/mL | - |
| Au nanorods [78] | PS 30, 200 nm PMMA 300 nm | PS 200 nm: EF = $10^2$ PS 30 nm: EF = $10^2$ (Equation (4)) | 532, 785 nm | PS 30 nm: 1.17 µg/mL | - |
| Device | | | | | |
| AuNPs on glass [84] | PS 161, 33 nm PET 62 nm | EF = 60–450 (Equation (3)) | 785 nm | 46 nm AuNPs: PS 33 nm: 20 µg/mL PS 161 nm: 10 µg/mL PET 62 nm: 15 µg/mL 14 nm AuNPs: PS 33 nm: 20 µg/mL PS 161 nm: 20 µg/mL PET 62 nm: 32 µg/mL | -- |
| Filter paper with AuNPs [94] | PET 10, 15, 20 µm | PET 10 µm: EF = 61.30 PET 15 µm: EF = 115.58 PET 20 µm: EF = 360.50 (Equation (3)) | 532 nm | PET 10, 15, 20 µm: 100 µg/mL | PET 20 µm in tap water and pond water: 100 µg/mL |
| Au on glass [79] | PS 100–800 nm | *p*-ATP: EF = $4.2 \times 10^8$ (Equation (2)) | 532 nm | PS 100 nm: 0.26 µg/mL PS 300 nm: 0.17 µg/mL PS 600 nm: 0.10 µg/mL PS 800 nm: 0.10 µg/mL | -- |
| Au on AAO nanopores [81] | PS 1, 2, 5 µm PMMA 1, 2 µm | PS 1 µm: EF = 5 PS 2 µm: EF = 20 PS 5 µm: EF = 6.5 PMMA 2 µm: EF = 8 PMMA 5 µm: EF = 3.5 (Equation (2)) (Single particle) | 785 nm | -- | -- |
| Au nanowires [86] | PS 100–800 nm | PS 800 nm: EF = 979 PS 300 nm: EF = 2219 (Equation (5)) | 638 nm | -- | -- |

**Table 3.** *Cont.*

| Substrate | Plastics | EF | $\lambda_{exc}$ (nm) | LOD (µg/mL) | Real Sample LOD |
|---|---|---|---|---|---|
| Klarite [88] | PS-PMMA 360 nm–5 µm | PS 360 nm: EF = 172<br>PS 500 nm: EF = 127<br>PS 1 µm: EF = 97<br>PS 2 µm: EF = 12<br>PS 5 µm: EF = 20<br>PMMA 360 nm: EF = 20–30<br>PMMA 500 nm: EF = 11–15<br>PMMA 2 µm: EF = 2–4<br>PMMA 5 µm: EF = 4–8<br>(Equation (2)) | 785 nm | -- | -- |
| AuNRs and AgNWs on cellulose [85] | PS 84–630 nm | AuNRs/RC CV: EF = $5.4 \times 10^6$<br>AgNWs/RC CV: EF = $1.8 \times 10^7$ | 785 nm | AgNWs/RC:<br>PS 84 nm: 100 µg/mL<br>PS 444 nm 50 µg/mL<br>PS 630 nm 100 µg/mL<br>AuNRs/RC:<br>PS 84 nm: 500 µg/mL<br>PS 444 nm: 500 µg/mL<br>PS 630 nm: 500 µg/mL | -- |
| Sputtered Ag on SiO$_2$ [80] | PS 200 nm–1 µm | *p*-ATP: EF = $2.3 \times 10^8$<br>(Equation (2)) | 633 nm | PS: 0.5 µg/mL | PS: 5 µg/mL in bottled, tap, and river water. |
| AAO, MoS$_2$ layer, and AgNPs [83] | PS 100–300 nm | R6G EF = $1.95 \times 10^{10}$<br>(Equation (2)) | 532 nm | R6G: $10^{-12}$ M | -- |
| AuNSs@Ag [87] | PS 400 nm–4.8 µm | -- | 633 nm | PS 400 nm: 50 µg/mL<br>PS 800 nm, 2.3, 4.8 µm: 100 µg/mL | PS 400 nm: 500 µg/mL in sea water, tap water, and river water |
| Ag/CuO NW in bowl shaped structure [89] | PS 20–900 nm | R6G: EF = $2.696 \times 10^8$<br>PS 20 nm: EF = 1107.7<br>(Equation (2)) | 532 nm | PS 20 nm: 0.1 µg/mL | -- |
| Detection and Remediation | | | | | |
| Ag nanowire array [67] | PS (50 nm to 1 µm) PMMA 500 nm | PS 50 m: EF = 238,096.10<br>PS 100 m: EF = $4 \times 10^4$<br>PS 300 m: EF = 840.36<br>PS 500 m: EF = 299.42<br>PS 1 µm: EF = 75.07<br>(Equation (2)) | PS: 785 nm PMMA: 633 nm | PS 50 nm: $10^{-4}$ µg/mL PMMA 500 nm: $10^{-3}$ µg/mL | PS 500 nm: $10^{-3}$ µg/mL in seafood, market water, and seawater. |
| AuNPs decorated sponge [74] | PS, PET, PE, PVC, PP, and PC (80–150 µm) | 4-MPY: EF = $1.36 \times 10^9$<br>(Equation (3)) | 785 nm | 4-MPY: $1.1116 \times 10^{-4}$ µg/mL PS: 1 µg/mL | PS: 50 µg/mL in seawater, river water, snow water, and rainwater. |
| Other approaches | | | | | |
| Au nanopore [69] | PS 20 nm | 4 Mbn: EF = $1.87 \times 10^5$<br>(Equation (2)) | 633 nm | -- | -- |
| AuNP100 nm [96] | PS 500 nm–10 µm PMMA 300 nm–1 µm | -- | 532 nm | -- | -- |

Abbreviations used in this table: Rhodamine 6G (R6G), p-aminothiophenol (*p*-ATP), 4-mercaptopyridine (4-MPY), 4-mercaptobenzonitrile (4 Mbn).

When using SERS for MNPs detection, three main strategies can be identified:

- SERS detection based on colloidal noble metal nanoparticles;
- SERS detection employing nanostructured devices;
- Hybrid SERS system capable of both the detection and remediation of MNPs.

We describe each approach in detail as well as the relevant literature in the following sections.

### 4.2.1. SERS Detection Based on Colloidal Noble Metal Nanoparticles

One of the most popular strategies for the SERS detection of micro- and nanoplastics is a direct method based on colloidal noble metal nanoparticles without requiring any substrate fabrication.

First of all, it should be emphasized that the LSPR, and in turn, the electromagnetic enhancement of the field, are strictly related to the nature and geometry of the SERS-active particles. In this regard, great efforts have been made in the last years to achieve high enhancement factors using different noble metal nanoparticles and, as will be discussed later, silver and gold nanostructures are the most widely used metals in SERS substrates due to their physical and LSPR properties [126].

Another important factor that strongly influences the SERS enhancement is the possible aggregation of colloidal nanoparticles, which could result in nanoparticle clusters with different plasmonic properties with respect to isolated NPs, and with enhancement efficiencies that exceed those of the latter.

In most of the works reported in this section, the detection of micro- and nanoplastics was made by adding salts and/or acids to colloidal systems, causing their precipitation and aggregation, thus increasing the SERS hotspots.

In 2020, Lv et al. [123] used this exact approach, with spherical silver nanoparticles (average diameter of $56 \pm 0.82$ nm, see Figure 3a) and sodium chloride as their aggregating agent [123]. Their paper describes the detection of three different types of MNPs by first making a comprehensive study of the best conditions in which to run the tests. The volume ratio of the PS sphere aqueous solution to silver colloid, NaCl concentration, and the concentration of the PS samples were optimized. The best result was obtained for the detection of PS spheres of 100 nm and 500 nm, for which a LOD of 40 μg/mL was reached in both pure water and seawater. The SERS signals of PE and PP plastic powder with a size of 10 μm were also recorded in both pure water and seawater, but the enhanced signal of these microplastics was not as strong as that of the PS nanoplastics, most likely due to the larger size of PE and PP. In summary, Lv et al. provided an efficient SERS substrate for the quantitative and qualitative detection of PS nanoplastics, which, however, showed limitations in the identification of PE and PP micropowder.

Another work based on the aggregation of silver nanoparticles was published by Zhou et al. in 2021 [68]. The authors used silver spheres with an average size of $56.7 \pm 14.1$ nm, and aggregated them with PS nanoplastics that were 1 μm and 50 nm in size using magnesium sulfate ($MgSO_4$) as the coagulant agent. Specifically, AgNPs were mixed homogeneously, then the PS aqueous solution and $MgSO_4$ and 5 μL of the resulting sample were placed on a clean silicon wafer and dried. SEM images were taken to investigate the morphology of the aggregates before analyzing them by SERS. A lower LOD than the previous work was reported, around 5 μg/mL for both the 1 μm and 50 nm PS particles. In spite of the interference peaks from the sample matrix, since detection was carried out n spiked river water samples, several characteristic PS peaks were obtained, demonstrating the power of SERS to probe nanoplastics.

Hu et al. [76] also worked with a similar system for quantitative nanoplastics analysis, in which potassium iodide was added to silver nanoparticles (50–60 nm), working both as an aggregating agent and as a cleaner to remove impurities from the surface of the silver nanoparticles, which is crucial for a quantitative investigation. Polystyrene nanoplastics of different sizes were detected in aqueous media, with concentrations ranging from 6.25 μg/mL to 2000 μg/mL, with LODs of 12.5 μg/mL for 50 nm PSNPs, 6.25 μg/mL for 100 nm PSNPs, 25 μg/mL for 200 nm PSNPs, and 25 μg/mL for 500 nm PSNPs. Compared to the previous works based on the aggregation of silver nanoparticles for plastics detection, the proposed method exhibited high sensitivity, with a LOD very close to the one in the work by Zhou et al. Good repeatability, linear relationships for all nanoplastic sizes, interference resistance, and quantitative analysis capacity were reported.

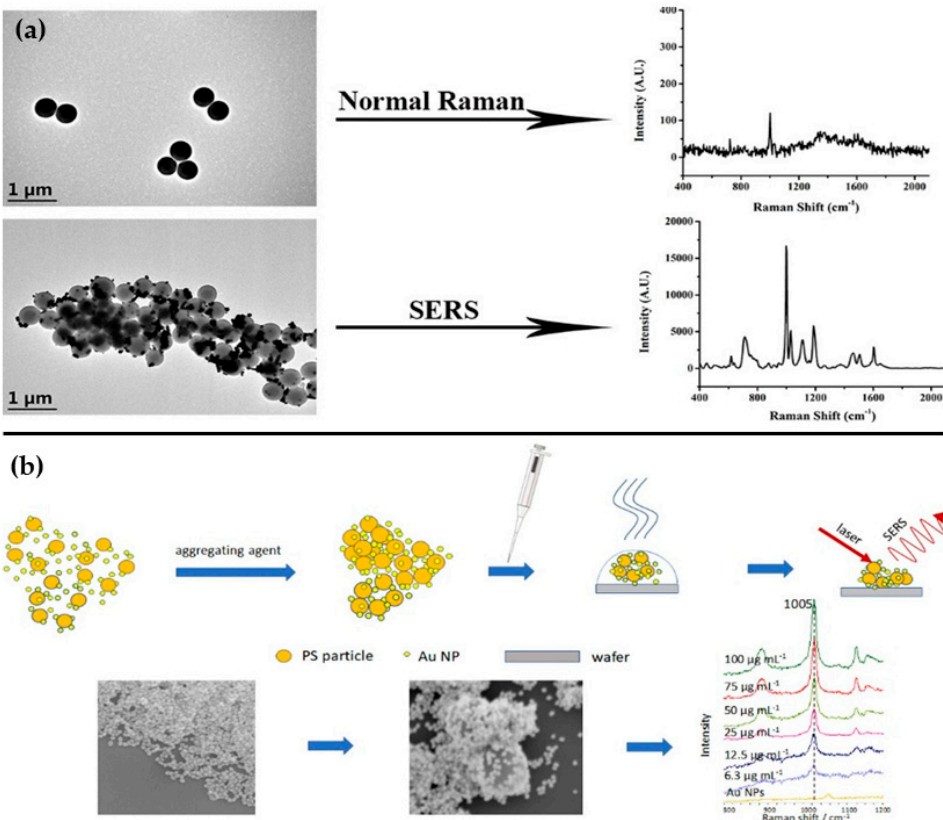

**Figure 3.** (**a**) Schematic representation of the work of [123], where AgNPs of 56 nm were used as the SERS base, inducing their aggregation with NaCl to further increase their enhancement effect. Plastic particles, in the size range from nanometers to microns, could be detected here, reaching the best results for 100 nm PSNPs with a LOD of 40 µg/mL. Reprinted from [123], Copyright 2020, with permission from Elsevier. (**b**) Schematic illustration from ref. [77] for the detection of 350 nm PSNPs and PE microplastics, starting from gold nanoparticles as the SERS base and then treated with different aggregating agents to boost their SERS effect. In the bottom row from left to right, the SEM images of the AuNPs and PSNPs with AuNPs are reported, concluding with SERS spectra of 350 nm PSNPs in different concentrations. Reproduced from ref. [77], Creative Commons Attribution 4.0 https://creativecommons.org/licenses/by/4.0/ (accessed on 12 August 2023).

Silver is not the only metal that can provide a good SERS effect: the use of gold with a similar particle aggregation approach has also been reported in the literature. In 2022, Kihara et al. [71] detected PS nanoplastics using a filter paper-based method with gold nanoparticles. Specifically, they relied on gold nanoparticles with an average size of $23.2 \pm 4.2$ nm, obtained via a reversed Turkevich synthesis, for the detection of polystyrene nanoplastics with different sizes (20 nm and 200 nm) and surface functionalization (carboxylic acid and amidine-modified, PS($-$) and PS($+$) respectively). Prior to deposition onto filter paper, AuNPs and PSNPs were pre-mixed in equal parts, then the colloidal mixture was added dropwise to filter paper and left to dry in air. This system allowed for the detection of positively charged 20 nm PSNPs up to 10 µg/mL, whereas for larger PSNPs($+$) (200 nm), the LOD was significantly reduced to 1 µg/mL. They also tested this substrate in the presence of NaCl to simulate the ionic strengths of biological fluids and seawater, lowering the LOD to 100 µg/mL for both 20 nm PSNPs($+$) and 200 nm PSNPs($+$). For the negatively charged PSNPs, on the other hand, no enhancement effect was reported. This is probably due to the electrostatic repulsion between PSNPs and AuNPs, both negatively charged, preventing PSNPs from being in the SERS hotspot of AuNPs.

In summary, this filter paper-based SERS system is an efficient method for the detection of 20 nm and 200 nm positively charged PSNPs. This paper is a good starting point for the

study of the detection of PSNPs in real samples. The same substrate could also be tested for negatively charged PSNPs by considering the use of positively charged AuNPs.

Mikac et al. [77] used a similar approach, starting from the synthesis of gold nanospheres of different dimensions (average sizes of 33 nm, 67 nm, and 94 nm) and gold nanorods with shorter and longer diameters of 23.5 nm and 35.5 nm (see Figure 3b). All of these nanostructures were subsequently tested for the detection of 350 nm PS nanoplastics to select the one with the best SERS response, which were the gold nanorods (AuNRs) with a LOD of 6.25 µg/mL. More tests were conducted to find the optimal conditions for the SERS measurements, from refining the volume ratio between the gold colloid and PS sample, to the choice of the aggregating agent and its optimal concentration. In contrast to earlier findings with 350 nm PSNPs, such promising results were not obtained for the polyethylene microplastics (1–4 µm), whose SERS signals were difficult to obtain, probably because the PE plastics were not completely covered with AuNRs.

In summary, the results of Mikac et al. are comparable with the data from Kihara et al. with gold nanoparticles. The same can be said for the results published with silver nanoparticles for PSNPs from Zhou et al. and Hu et al. The major limitation of the former work is the lack of detection in real samples.

In addition to the works based on the aggregation of gold nanoparticles with MNPs by the addition of salts, two other papers [78,125] reported on NP aggregation achieved through different approaches.

Park et al. [125] worked with on-site 3D assemblies of colloids ranging from nano- to microparticles for sensitive SERS detection without specific binding by exploiting the photothermally driven convective flow. With this method, the authors obtained co-assemblies of CTAB-stabilized gold nanorods (10 nm × 36 nm) with larger PS microparticles of various sizes (0.1, 0.5, and 1 µm). Concentration-dependent SERS spectra were obtained for all sizes of PS particles, measuring a LOD of 1 µg/mL for 0.1 µm PS particles, one of the lowest presented in this section. They also conducted a SERS characterization of the substrate using Rhodamine 6G as the probe molecule, resulting in an enhancement factor of approximately $5.4 \times 10^5$, which was higher than those in the other previous works.

The second work is the one of Yu et al. [78], where they carried out a rapid in situ Raman detection of PS nanoplastics based on dielectrophoresis (DEP) and AC electro-osmosis (ACEO), known as DEP-ACEO-Raman tweezer (DART).

This technology, outlined in Figure 4a, allows one to obtain Raman signals in real-time from PS-underwater nanoplastics with ultralow concentrations, exploiting a field-induced active collection of 30 nm PSNPs and gold nanoparticles toward the optical sensing area of the DART substrate from remote areas. This system, besides a rapid aggregation, allows one to simultaneously achieve the preconcentration, separation, identification, and on-site detection of target molecules. Moreover, thanks to the addition of plasmonic nanostructures that support Raman amplification, this strategy can overcome the diffusion limits and low Raman scattering cross-section of nanoplastics.

With this method, for 30 nm PSNPs, the lowest LOD of all the devices (0.00117 µg/mL) based on colloidal NP detection was obtained, even lower than the works with silver nanostructures, which are known to have a higher SERS amplification effect than the gold ones. In addition, tests were also carried out on the target-selective clustering and separation of two different nanoplastics by using a mixture of PMMA-NPs and PSNPs: SERS spectra of PS and PMMA particles were individually registered by applying two different AC signals by the lapse of time.

In summary, the last two works by Park et al. [125] and Yu et al. [78] were the two with the lowest LODs in this paragraph, and they both reported a SERS characterization of their respective gold nanorods and an evaluation of their enhancement factors, lacking instead in detection tests on real samples.

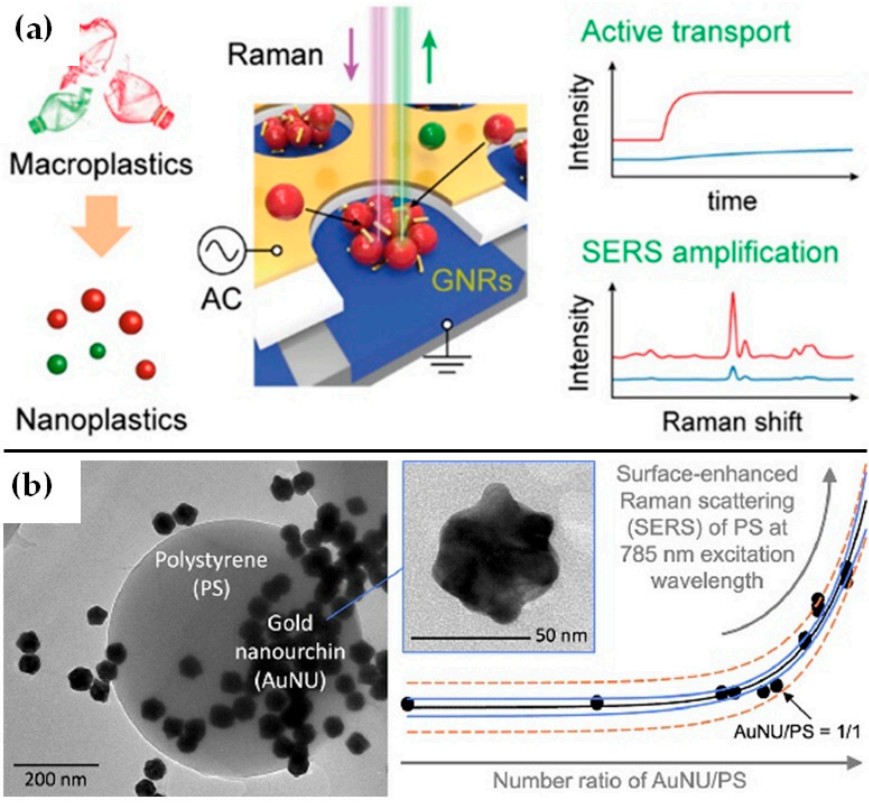

**Figure 4.** (**a**) Schematic representation of work from ref. [78]: Raman signals in real-time were measured from underwater nanoplastics via AC electro-osmotic flows and dielectrophoretic tweezing, combined with plasmonic nanoparticles, accomplishing the highest on-site detection performance. Reprinted with permission from ref. [78]. Copyright 2022, American Chemical Society. (**b**) Schematic illustration of the paper [124]: the effectiveness of SERS on PSNPs was evaluated at a single-particle level with a different number of gold nanoparticles to determine the minimum conditions required for the SERS effect. Reprinted from [124]. Copyright 2022, with permission from Elsevier.

Having discussed colloidal substrates based on the aggregation of noble metal nanoparticles, the final section of this paragraph addresses the only work based on simple gold nanoparticle colloids [124]. Lee and Fang used 50 nm gold nanourchins (AuNUs) as the SERS substrate, and 100 nm PS nanoplastics as the probe analyte. The goal of this work was to evaluate the minimum number of AuNUs required for PS detection at a single-particle level, instead of determining the limit of detection of the PSNPs (see Figure 4b). The PS concentration was kept constant, while the concentration of AuNUs in terms of particle number was varied. Samples were then air-dried on a flat surface of aluminum and the SERS spectra were collected, recording an increase in the intensity of PS signals as the number of AuNUs rises. However, this work does not provide information on the SERS characterization of the substrate and the limit of detection of PS nanoplastics, neither in the pure water nor real samples.

The last paper to be discussed in this section is actually on the borderline between this paragraph and the next on the SERS detection of micro- and nanoplastics employing nanostructured devices. Pang et al. [82] were able to detect 100 nm PS nanoplastics at a concentration of 200 μg/mL by using spherical gold nanoparticles with an average diameter of 62 nm. For the SERS measurements, the substrates were prepared just by drying AuNP drops on a silicon wafer. The PSNP samples with different concentrations were placed on the substrate. The authors tested different concentrations of PSNP aqueous solutions ranging from 1600 μg/mL to 200 μg/mL. Besides determining the LOD, they also registered a strange behavior in the SERS spectra of the PS samples with concentrations in the range of 1000–1600 μg/mL. Despite the increase in PS concentrations, the intensity

of their SERS signals gradually decreased, probably due to the excessive number of PS particles adsorbed on the substrate.

### 4.2.2. SERS Detection Employing Nanostructured Devices

The second main strategy for the detection and quantification of MNPs is based on the use of highly sensitive SERS-active devices. The general idea is to develop a reproducible nanoplatform based on a SERS-efficient metal substrate. Over the last few years, several research groups have developed many SERS-substrates employing very different synthetic approaches and starting materials, resulting in promising sensitive devices for plastic pollutants. Different shapes and sizes of pure noble-metal nanomaterials have been investigated and employed with very promising results.

A good example of this approach is the functionalization of glass slides with spherical gold nanoparticles (AuNPs), creating a simple SERS substrate [84]. After being cleaned in piranha solution, glass microscopy slides were soaked in poly allylamine hydrochloride solution, rinsed with water, and finally soaked in a AuNP dispersion for 4 h, resulting in Au-functionalized glass slides. Both 14 and 46 nm AuNPs were synthetized to obtain the substrate and to carry out the detection tests for MNPs. The above-mentioned device was capable of detecting both PS and PET nanoparticles, reaching the lowest LOD of 10 µg/mL by employing 46 nm AuNPs to detect 161 nm of PS nanoparticles [84]. A SEM image of the 161 nm PS nanoparticles on the gold substrate is reported in Figure 5a.

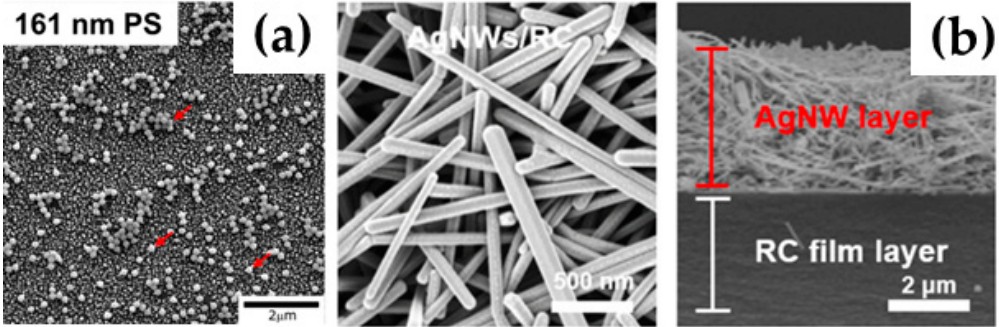

**Figure 5.** (**a**) Representative SEM image of the 161 nm PS nanoparticles after their addition on the glass slides functionalized with AuNPs, adapted from [84]. Examples of plastic particles are highlighted with red arrows. Creative Commons 4.0 https://creativecommons.org/licenses/by/4.0/ (accessed on 12 August 2023. (**b**) FE-SEM images of AgNWs/RC film (surface density 0.99 mg/cm$^2$, thickness 3.6 µm) on the left and cross-sectional FE-SEM images of the same substrate on the right. Adapted from [85]. Copyright 2021, Elsevier Ltd. All rights reserved with permission from Elsevier.

Xu et al. [94] also employed AuNPs with an average diameter of 23 nm as a hotspot source. Filter paper supported AuNPs were used to capture and detect PET microplastics. The substrate preparation is very simple: 80 µL of gold colloidal solution is dropped onto a square of filter paper, then dried in an oven at 35 °C. The pores of the fiber filter paper could trap the PET microplastics employed in this study (10, 15, and 20 µm), allowing their identification down to concentrations of 100 µg/mL. Real sample analyses were carried out to find out whether the filter paper doped with AuNPs was capable of detecting microplastics in real water samples. Interestingly, with spiked PET microplastics of 20 µm, the same LOD obtained in standard solutions of microplastics (100 µg/mL) was achieved, making this device suitable for the analysis of real samples.

Chaisrikhwun et al. [79] reported another gold SERS-substrate, but in this case, it was obtained by sputtering a thin layer of gold on glass slides. This method is fast, cheap, and simple, and the obtained device is capable of detecting nanosized PS particles of 100, 300, 600, and 800 nm, with a very low detection limit calculated to be around 0.10 µg/mL. The enhancement factor employing *p*-ATP as a Raman probe (reported in Table 3) was evaluated to identify the best thickness of sputtered gold, which proved to be 32 nm.

Other papers have reported the use of gold in different applications and methods to achieve a more reproducible and robust substrate. Shorny et al. [86] employed a 3D crossed gold nanowire substrate that enabled the detection of PS particles smaller than 1 μm. Another reported example is the use of a Klarite substrate, an ordered, dense grid structure of cavities (or "pits") with an inverted pyramid shape [88] capable of detecting nanoplastics down to 360 nm. With different EF, both the PS and PMMA particles could be identified, but it should be mentioned that to achieve the best Raman signal enhancements, the location of the plastics onto the substrate is critical. Being aware of this, when plastics reach the center of the pits, distinct Raman peaks are displayed.

Although it is possible to obtain ordered and homogeneous structures with gold, it is well-known that silver provides the highest SERS enhancements in the visible range. A lot of interest has thus been dedicated to silver-based SERS devices.

The work of Jeon et al. [85] compared the SERS signals of regenerated cellulose (RC) hydrogel films with AuNPs and silver nanowires (AgNWs). As expected, it can be observed from the EF listed in Table 3 that the AgNWs substrate was more sensitive than the one with AuNPs. This was also confirmed by the detection limits achieved from both substrates for the identification of PS nanoparticles. With the gold substrate, the characteristic Raman peak at 1002 cm$^{-1}$ was visible at a concentration of 500 μg/mL for all three different PS studied (84, 444, 630 nm). On the other hand, for the silver substrate, nanoplastics could be detected at a concentration of 100 μg/mL, confirming silver's greater ability to enhance the weak Raman signals.

Furthermore, to improve the SERS performance, many researchers have managed to prepare highly ordered noble metal substrates to achieve a greater homogeneity and a more densely packed arrangement of metal nanostructures. A popular support for the construction of SERS devices is anodic aluminum oxide (AAO), chosen mainly for its highly uniform nanoporous structure. Liu et al. [81] obtained V-shaped nanopores on AAO with deposited AuNPs through magnetron ion sputtering. This approach generated several hotspots with which PS and PMMA microplastics have been successfully identified by SERS. Using 1, 2, and 5 μm microplastics as the target, the obtained EFs allowed for some interesting observations. First of all, the Raman signals of the PS microspheres obtained with both ion and magnetron sputtering, compared with the silicon wafer, were significantly enhanced due to the presence of AuNPs. Moreover, with both PS and PMMA microspheres, it was possible to observe the single-particle Raman signal, proving that a AuNP@V-shaped AAO SERS substrate is suitable for the detection of different plastics. According to the calculation results of EFs reported in Table 3, the strongest SERS signals were exhibited when the magnetron sputtering method was employed to prepare the SERS substrate, probably because it provides a more uniform "hotspot" array. It must also be pointed out that the AuNP@V-shaped AAO SERS substrate was more sensitive for PS than for PMMA, according to the weaker Raman cross section of the latter.

With the common aim of obtaining internal hot spots in the AAO structure, Lê et al. [87] inserted silver-coated gold nanostars (AuNSs@Ag) into the nanopores by ultrasonication-induced self-assembly. The anisotropic shape of the nanostars enables a great SERS enhancement and makes this nanoplatform suitable for MNPs detection. The AuNSs@Ag dimer arrays exploited high homogeneity in self-assembling into the AAO nanopores and the SERS substrate obtained was employed to study the detection of PS nanoparticles of assorted sizes in several concentrations in distilled water, as represented in the SEM images reported in Figure 6a. Thanks to the nanosized gap, when the small nanoplastics interact with the branches of the AuNSs@Ag, a great SERS enhancement is achieved, if compared with the bare AAO structure. In this work, Lê et al. investigated the size-dependent detection of PS in an aqueous environment with different concentrations. At 10 mg/mL and 1 mg/mL, all of the different investigated dimensions of plastics (400 μm, 800 μm, 2.3 μm, and 4.8 μm) were observed, while only the 400 μm PS nanoplastics showed a detectable Raman signal at the concentration of 50 μg/mL. These interesting results revealed that SERS detection should depend on the size of the PS MNPs. To complete the study, MNPs

detection in the environmental water sample were also performed in order to quantitatively monitor PS particles via SERS. The tap, river, and seawater samples were analyzed reaching 500 µg/mL as the limit of detection for 400 µm PS nanoplastics.

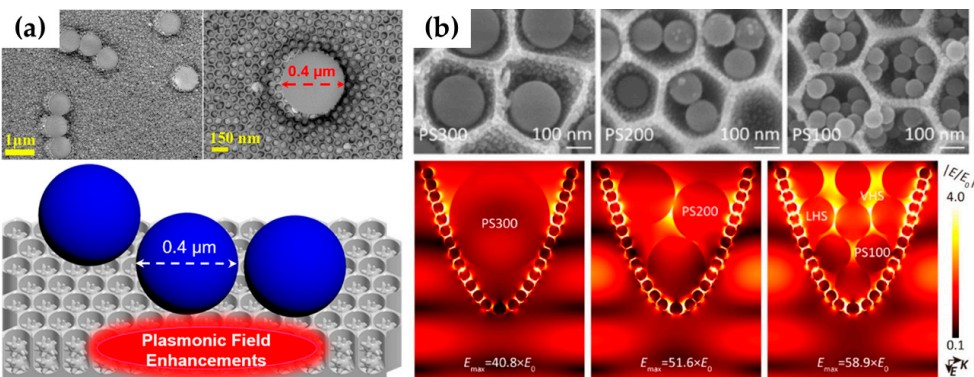

**Figure 6.** (**a**) SEM images of 400 nm PS nanoplastics (0.1% concentration) on the AuNSs@Ag@AAO substrates (**top**) and a schematic representation of the plausible SERS detection mechanism of nanoplastics with the AuNSs@Ag@AAO substrate (**bottom**), Reprinted from [87]. Copyright 2020, Elsevier B.V. All rights reserved with permission from Elsevier. (**b**) SEM images of AAO/MoS$_2$/Ag containing PS nanoplastics (**up**) and the simulated electric fields in AAO/MoS$_2$/Ag containing PS nanoplastics of different sizes. Adapted with permission from [83]. Copyright 2022, American Chemical Society.

This particle-in-cavity (PIC) structure was also employed to develop an AAO conical orifice with a 2D MoS$_2$ layer and 15 nm AgNPs [83]. Li et al. exploited the capacity of PIC structures to obtain a wide distribution of hotspots from inside the dielectric cavity to close to the sidewall, in order to detect water contaminants from ions to nanoplastics. More specifically, the substrate was composed of an AAO conical orifice, AgNPs, and a MoS$_2$ interlayer between the AAO and the AgNPs as both a chemical enhancer and internal standard for the PIC architecture. Employing R6G as the Raman probe, the authors found that Ag with a thickness of 15 nm had the highest intensity of the characteristic Raman peak. The limit of detection was achieved with a $10^{-12}$ M solution of R6G, with a resulting EF equal to $1.95 \times 10^{10}$. Beyond ions and molecules, this substrate also proved to be advantageous, thanks to the obconic cavity, for the detection of nanoplastics. Due to the curvature of cavities, MNPs inside the pores are entirely surrounded by AgNPs, thus enabling their recognition, which was confirmed by experimental tests of 100, 200, and 300 nm PS nanoparticles (see Figure 6b). Li et al. reported Raman spectra where the characteristic Raman peak of PS at 1002 cm$^{-1}$ was observed, even though they did not specify the concentration at which the spectra were acquired.

Alongside the use of AAO-based structures, other interesting SERS devices have been proposed in the literature. Chang et al., for example, developed a NWERS (nanowell-enhanced Raman spectroscopy) substrate composed of self-assembled SiO$_2$ sputtered with a silver film (SiO$_2$ PC@Ag) [80]. Their strategy was to exploit the coffee-ring effect and the gravity of nanoplastics to trap them exactly on the nanowells, thus enabling their detection in water environments. Through a simple two-step approach, the SiO$_2$ PC@Ag substrate was formed: first, using the interfacial assembly method, the structured SiO$_2$ nanospheres assemblies were fabricated, then silver was sputtered to obtain a thin shell to acquire the SERS effect. To optimize the SERS performance, the thickness of the sputtered silver was studied by observing the enhancement of $p$-ATP. With a shell thicker than 100 nm, the inter-nanosphere gaps disappeared, consequently, an 80 nm silver shell (SiO$_2$ PC@80 nm Ag) was chosen as the best substrate (see SEM images in Figure 7a). Prior to studying the performance of this substrate to detect PS MNPs, Chang et al. collected different Raman spectra by employing three different laser sources, finally choosing a 633-nm laser as it matched the photonic bandgap and possessed better SERS performance.

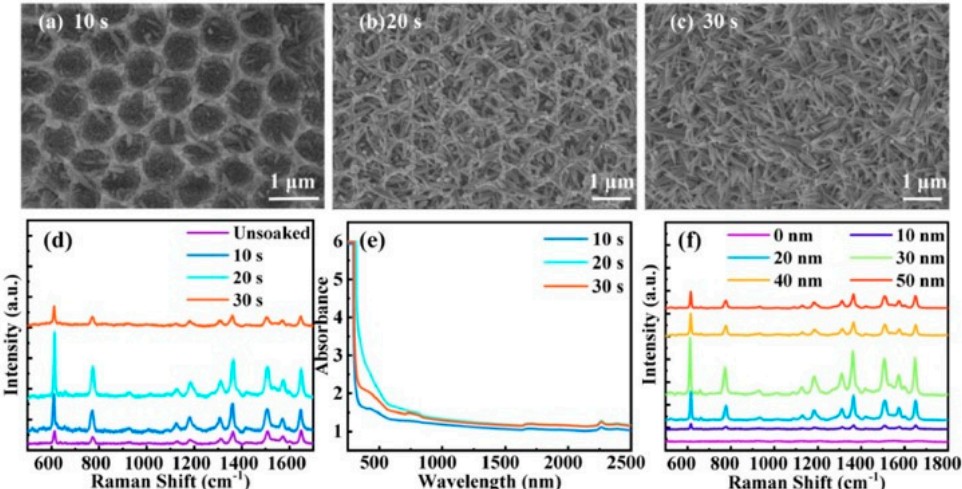

**Figure 7.** SEM images of the Cu/BaTiO$_3$@PVDF bowl-shaped substrates after soaking in NaClO solution for different times in order to obtain CuO NWs: subfigures (**a**–**c**) show samples prepared at 10s, 20s and 30s times, respectively. (**d**) Raman spectra of R6G 10$^{-6}$ M on the SERS substrate after soaking in NaClO solution for different times (unsoaked, 10, 20, and 30 s). (**e**) UV–Visible spectra of the Cu/BaTiO$_3$@PVDF bowl-shaped substrates after soaking in NaClO solution for different times (10, 20, and 30 s). (**f**) Raman spectra of R6G on the Ag/CuO NW/BaTiO$_3$@PVDF bowl-shaped substrate with different thicknesses of the Ag layer. Reproduced from [89]. Copyright 2023, Optica Publishing Group under the terms of the Optica Open Access Publishing Agreement.

PS nanospheres with sizes of 200, 500, 800 nm, and 1 μm were analyzed on the optimized SiO$_2$ PC@80 nm Ag. Despite the particle size, the LODs all reached 0.5 μg/mL (0.00005%) in deionized water, observing that the intensity of the characteristic of PS (1003 cm$^{-1}$) increased with the concentration until it reached 0.1%. The analysis of real samples was also performed, reaching a 5 μg/mL LOD with spiked PS nanoplastics in bottled water, tap water, and river water.

With the common purpose of producing a SERS device capable of detecting nanoplastics with the help of cavity-structure, Lv et al. developed a nanowire-in-bowl-shaped piezoelectric cavity structure [89]. In particular, the bowl-shaped structure was built on a BaTiO$_3$@PVDF film and then CuO nanowires (NW) were formed inside it, obtaining a CuO NW/BaTiO$_3$@PVDF bowl-shaped structure. In addition, to confer SERS activity to the substrate, 30 nm of silver was deposited after evaluating the variation in the SERS activity with R6G by varying the silver thickness, as can be observed in Figure 7f. The main advantage of this device is the optical field amplification, thanks to the large specific surface area and light focusing properties of the bowl-shaped structure. Additionally, the dense CuO NWs into the structure create a multi-dimensional SERS hotspot, thus achieving an excellent enhancement factor, equal to 2.7 × 10$^8$. Moreover, being able to bind the nanoplastics, this bowl-shaped substrate was employed to successfully detect PS nanoplastics up to 20 nm in size. Compared with larger MNPs (100 and 300 nm), the 20 and 50 nm PS nanospheres showed higher Raman intensity, demonstrating that this SERS substrate had directional detection ability for smaller sized nanoplastics.

Beyond the classical so-called SERS chips, in recent years, some other interesting devices for MNPs detection have been reported in the literature. For example, a wearable SERS sensor based on ultrathin, flexible biointegratable gold nanomesh [96]. This particular device was demonstrated to be highly scalable, easy to fabricate, low cost, flexible, and adhesive, very important features for a wearable sensor prepared with the purpose of detecting different substances such as microplastics.

This sensor is made of gold-coated biocompatible polyvinyl alcohol (PVA) nanofibers capable of sticking onto virtually any surface such as human skin. In this work, Liu et al. optimized the dimension of the substrate in order to obtain SERS capabilities while

maintaining the features listed above. The number of hotspots in a unit volume was optimized by decreasing the diameter of the nanowires, and the final setup was reached by selecting 490 nm for the nanowire diameter and a deposited gold thickness of 150 nm. For a proof-of-concept, by employing R6G as the Raman probe, an enhancement factor of 108 was obtained, and a $10 \times 10^{-9}$ M concentration of R6G was the lowest detectable. Then, to demonstrate the capability of the sensor for health monitoring, Liu et al. successfully detected human sweat biomarkers (urea and ascorbic acid) and some abuse drugs. Besides the detection of molecules of biological interest, polyethylene microplastics in water were successfully identified at a concentration of 0.1%, but no other tests were performed to determine the LOD. Despite that, these first results show that the device is robust for microplastics detection both in dry and wet conditions with a very high potential for in situ monitoring.

Another interesting method that exploits the SERS effect for nanoplastics detection was proposed by Nie et al.: a single gold nanopore was fabricated on the tip of a glass nanopipette as the supporter [69]. To fabricate this interesting device, first, a cap-like gold structure was achieved by in situ reduction to cover the glass tip. Then, they employed a high voltage electric pulse to create a single nanopore in the gold cap by optimizing the drilling condition to obtain a diameter ranging from 30 to 50 nm and employing a 0.1 mM KCl solution as the nanopore drilling solution. To evaluate the SERS activity of this device, 4-mercaptobenzonitrile (4 Mbn) was employed as the Raman probe obtaining an enhancement factor of $1.87 \times 10^5$. The results obtained by testing the 20 nm PS nanoplastic solutions indicate that PS nanoplastics randomly pass through the nanopore, but with high variability on the Raman intensity. In addition, 10 nm PMMA nanoplastics were also identified. This gold nanopore device shows promising results for the qualitative and quantitative determination of nanoplastics, but a deeper study is needed to determine the limits of detection and the viability of the method on real samples.

### 4.2.3. Hybrid SERS System Capable of Simultaneous Detection and Remediation of MNPs

As mentioned in the previous paragraphs, SERS has attracted considerable attention in the last years for chemical sensing and environmental monitoring. With sensitivity up to the single molecule level and capability in the identification of molecules through their vibrational fingerprint, SERS is widely used to identify low concentrations of numerous chemical substances such as pollutants, biomolecules, and micro- and nanoplastics [77].

However, the detection limits of SERS are far from environmentally relevant concentrations [76], furthermore, SERS detection on targets without a specific adsorption group that do not present an affinity for the metallic nanostructure remains a significant challenge [127,128]. Another limitation of this technique is the lack of selectivity for the analytes when found in complex matrices: it is therefore necessary to develop a method to separate and pre-concentrate micro- and nanoplastics.

The two papers reported in this paragraph used membrane filtration for the pre-concentration, one of the most common methods in this regard.

For the detection of microplastics in non-pre-treated water samples, Yin et al. [74] used a sponge to locally concentrate microplastics, functionalizing it with a layer of 20 nm Au nanoparticles as the SERS substrate.

The sponge (commercial, made of styrene butadiene latex) was used as a template to assemble AuNPs, and its adjustable size could allow for the distance between the hotspots on the substrate surface to be controlled. This makes it possible to assemble AuNPs with improved gaps and avoid NP aggregation, producing a more uniform SERS substrate.

Moreover, the pores of the sponge allowed water to be rapidly separated from the sample, while their bowl-shaped structure effectively trapped microplastics (see the scheme in Figure 8a).

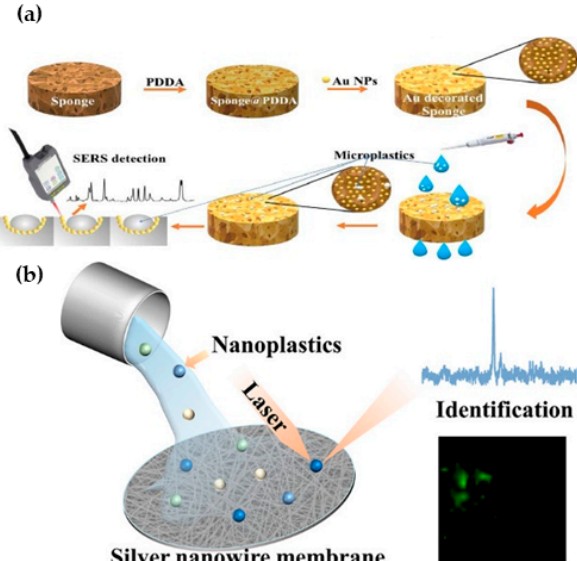

**Figure 8.** Panel (**a**) is a schematic representation of the work in ref. [74] with the synthesis of gold-modified sponges and the subsequent capture, pre-concentration, and SERS detection of microplastics. Reprinted from [74]. Copyright 2021, with permission from Elsevier. In the second part (**b**), the graphical abstract of Yang et al. [67] is reported, schematizing the preconcentration and analysis of nanoplastics using their hyphenated method of pre-concentration and detection using membrane filtration and SERS. Reprinted with permission from [67]. Copyright 2022, American Chemical Society.

The fabrication of the substrate is very simple: first of all, sponges were functionalized with PDDA (Poly(diallydimethylammonium chloride)), that is, the linkage between the sponge and AuNPs; after rinsing, sponges were soaked in the AuNP solution to be uniformly functionalized.

A preliminary study was made to optimize the substrate, fabricating it with one to eight AuNP layers: to evaluate their SERS response, crystal violet (CV) was used as the probe molecule, registering an increase in the intensity of the CV characteristic peak between one and five layers, and a decrease beyond five layers. Hence, the substrate with five layers of AuNPs was used for subsequent measurements.

Moving to the detection of plastics, the authors prepared aqueous solutions of different microplastics: polystyrene from Petri dishes, polyethylene terephthalate, polyethylene, polyvinyl chloride, polypropylene, and polycarbonate, all of them with dimensions of 80–150 μm.

First, they tested their substrate with PS-microparticles, noting that when the amount of the test solution changed, the amount of microplastics attached to the substrate also changed: for test solution volumes of 2 mL, 4 mL, and 6 mL, respectively, LODs of 50.0, 5.0, and 1.0 μg/mL were found. To test the selectivity of this substrate, they also exposed it to the other microplastics, noting that all their signals did not overlap and that, therefore, the signal at 1002 cm$^{-1}$ was selective and characteristic of PS-microplastics. In addition, analyses of real samples were also made, specifically in river water, seawater, rainwater, and snow water, where a LOD of 50 μg/mL was obtained.

The second paper [67] reported a silver SERS substrate. More specifically, the authors worked with Ag nanowires with a diameter of approximately 60 nm and a length of about 15 μm. The AgNWs were incorporated in a membrane prepared with the flow-through method: a commercial filter paper was connected to a syringe containing the AgNW solution that was flowed through the same filter paper under the action of pressing. The use of membranes to filtrate and separate micro- and nanoplastics is widely popular thanks to its numerous advantages (high efficiency, easy operation, maintenance of the original morphology of the particles, and the ability to retain particles of different sizes).

AgNW membranes were tested with 500 nm PS nanoplastics (see Figure 8b), showing increasing retention rates with the increase in the loaded volume of AgNW solution. Under the optimal conditions, 98.9% of PSNPs was trapped in the membrane. Some tests were also conducted with PSNPs of other dimensions (50 nm, 100 nm, 300–1000 nm), which always obtained a retention rate above 86%, suggesting that the AgNW membrane is suitable for enriching PSNPs of different sizes in water.

Before the detection of nanoplastics, the SERS performances of the AgNW membranes were tested using 4-aminothiophenol (PATP) as the probe molecule, evaluating this system for its:

- Sensitivity, by comparing the SERS spectra of PATP solutions with different concentrations;
- Time stability, by comparing the SERS spectra of samples newly prepared with those of one-day and one-week-old samples;
- Uniformity, where the AgNW membranes showed a good relative standard deviation of 3.68%;
- Laser irradiation stability, by using $10^{-6}$ M PATP, resulting in a relative standard deviation of 7.18%.

Another significant aspect of this work was the use of potassium iodide to clean the background signal of the SERS substrate: iodide helps to reduce the interference signals of the AgNW membrane, being a competitive adsorption agent for removing impurities on the surface of silver nanostructures. After the treatment with KI, the characteristic SERS peaks of the plastics were stronger, whereas on the other hand, the signals of PVP and the other impurities disappeared. The amount of KI was another important parameter that has to be evaluated, since it affects the enhancement performance of the substrate.

In the last part of the paper, they focused on the detection of nanoplastics; in more detail, they worked with PSNPs of different dimensions (50 nm, 100 nm, 300 nm, 500 nm, and 1 μm) and 500 nm PMMA-NPs: for 50 nm PSNPs, they registered a LOD of $10^{-4}$ μg/mL, while for 500 nm PMMA-NPs, a LOD of $10^{-3}$ μg/mL was obtained.

The detection of nanoplastics in environmental water samples was also tested, specifically working with seafood market water, seawater, and river water. The 500 nm PSNPs, with a concentration of $10^{-3}$ μg/mL, were detected in seafood market water and seawater, while in river water, the same concentration was not detectable.

In summary, both the hybrid SERS systems reported here resulted in very low LODs, but Yang et al. [67], who worked with silver nanostructures that are known to have a higher SERS effect than the gold ones, obtained the best result, reaching a LOD of $10^{-4}$ μg/mL for 50 nm PSNPs, the lowest of all the works examined in this review.

### 4.2.4. Influence of MNP and Plasmonic Nanomaterial Size on SERS Response and Enhancement Factor

It is interesting to observe that, despite the use of the same SERS substrate/colloid approach in most of the papers previously cited, different MNP sizes influenced the SERS response. As can be seen from Table 3, several of the listed papers reported the quantitative analysis of MNPs of different sizes with different results in the related EFs and LODs. For example, Hu et al. [76] studied the SERS enhancement of PS particles in a range from 50 nm to 500 nm by applying different SERS strategies. The best LOD value was obtained for the 100 nm sized PS, while PS particles with a larger size showed an extremely lower intensity of the characteristic Raman peak (1002 cm$^{-1}$) compared to the intensities obtained on smaller nanoplastic particles. Recalling that the electromagnetic enhancement is practically negligible for distances above 10–20 nm, this behavior can be explained since micrometer-sized particles possess a smaller specific surface area than nanometric ones; in fact, only a slight surface contact between microplastics and the SERS active substrate (here AgNPs) occurs, resulting in a poor SERS response.

Clearly, not only is the size of the target plastics important for obtaining a strong signal, but also the size, shape, and material of the plasmonic nanoparticles. The crucial importance of using colloids of appropriate shape and material has already been discussed

in the previous paragraphs. In brief, sharper morphologies can increase the presence of hotspots (consequently leading to higher EFs) as well as the use of silver (as the core nanomaterial or as a coating on different metals), which can scatter light more efficiently than gold. Regarding the size of the metal nanoparticles, it can be observed, for example, in the work of Caldwell et al. [84], that using SERS substrates with smaller (14 nm) or larger (46 nm) gold nanoparticles generates a different signal intensity. In fact, by employing 46 nm AuNPs, PS and PET nanoplastics could be detected at lower concentrations compared to the use of 14 nm AuNPs. If the plasmonic particles employed are indeed too small, the electronic scattering on the surface decreases, and consequently, the radiated field. Of course, it needs to be kept in mind that there is also an upper limit in dimensions, otherwise, the localized surface plasmons cannot be excited.

While SERS is a relatively recent application, its use for the determination of micro- and nanoplastics is still at a primordial stage. If what can be observed is a tendency for the size of the plastics to increase as the SERS response decreases, more study and investigation is still needed on the subject. Certainly, the difference between a colloidal and a substrate application, the attractive and repulsive forces between plastic pollutants and plasmonic particles, the influence of the sample preparation method, etc. are key points on which much work is still needed.

### 4.3. Challenges, Current Limitations, and Future Perspectives for the Use of Plasmonic and SERS-Based Techniques

The use of SERS is one of the most recent and promising developments in the field of micro- and nanoplastics detection.

This approach was tested in order to overcome the limitations encountered with other analytical techniques such as difficult sample pre-treatment, expensive and time-consuming analysis, etc.

The combination of the diagnostic Raman fingerprints of different polymers and the great enhancement of signals thanks to the proximity of plasmonic nanostructures makes SERS an interesting and viable replacement to other techniques that cannot usually provide simultaneous quantitative and qualitative information on MNP samples.

Moreover, the most promising prospects, while obviously still needing a lot of research, could be the in situ quantification (exploiting the use of portable Raman devices) and continuous analysis, which combine the use of fiber optics with the use of microfluidic circuits and real-time deep learning data processing.

The goal looks promising, as do the results achieved in recent years, but there are still gaps to be filled and technical issues to be resolved. One of the most critical aspects lies in the lack of a standard methodology for sample preparation and a standardized procedure of analysis. Another key aspect is the sample matrix. The main matrix in which MNPs can be found is definitely aqueous, which is why almost all of the proposed applications have focused their testing on aqueous samples. In recent years, many research groups have tested the effectiveness of their substrates on real water samples (sea, river, tap water. . .) spiked with MNPs (see Table 3). However, almost all of the above studies showed the response of their devices in the presence of only one type of plastic and one size at a time, clearly not reflecting what a real sample might be. Here, a new approach to allow for the simultaneous detection of different sizes and composition of MNPs is required to achieve a universal method to characterize a real unknown sample.

Regarding the role of plasmonic and SERS-based techniques in the complex landscape of MNPs detection, it is difficult to make accurate comparisons in this field. MNPs detection and quantification, indeed, still lacks recognized protocols and universal standards. So far, the legislation in the field is very limited. The most commonly used techniques are summarized in Table 1, together with their main limitations and the LODs when available. Plasmonic and SERS-based techniques could overcome some of these limitations, alone or in combination with other techniques.

## 5. Conclusions

The presence of micro- and nanoplastics in the environment is ubiquitous and unquestionably dangerous. Among the many different analytical techniques used for MNPs detection and quantification, plasmonic and SERS-based approaches are emerging as promising in the scientific literature. In this review, we tried to summarize all of the relevant papers recently published in this field, outlining the most interesting aspects, and possibly, the aspects that still need improvement. Plasmonic materials can be exploited in different ways: MNPs can be detected through a simple colorimetric assay, using a plasmonic chip (already available for other analysis), or by exploiting the peculiar fluorescence properties of the material. Among these approaches, however, the most promising technique seems to be SERS. In this case, the presence of a plasmonic material is crucial in order to enhance the weak Raman signal of the plastic analytes, allowing one to detect MNPs, and in some cases, even discriminate between the different polymers. This field, however, is still at its dawning. As we stressed in the last paragraph, the majority of the published studies has focused on ideal samples, and a complete evaluation of complex matrices and real samples is rarely found in the literature. This can be justified by the fact that most of the papers cited herein were published very recently (2020–2023). We thus believe that these proof-of-principle studies will soon be followed by other papers analyzing a wider application of these analytical techniques.

**Author Contributions:** Conceptualization, G.D. and A.T.; Writing—original draft preparation, G.D., S.S., M.P. and B.A.; Writing—review and editing, B.A. and A.T.; Visualization, S.S. and M.P.; Supervision, A.T. and P.G.; Funding acquisition, A.T. and P.G. All authors have read and agreed to the published version of the manuscript.

**Funding:** This research was funded by MIUR, PRIN 2017 project 2017EKCS35.

**Institutional Review Board Statement:** Not applicable.

**Informed Consent Statement:** Not applicable.

**Data Availability Statement:** No new data were created or analyzed in this study. Data sharing is not applicable to this article.

**Conflicts of Interest:** The authors declare no conflict of interest.

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
