# Peer review of "Plasmonic Nanomaterials for Micro- and Nanoplastics Detection"

_applsci, doi:10.3390/app13169291_

Round 1
Reviewer 1 Report
To enhance the article's overall impact, I suggest the following minor revisions:
-
Expand on the advantages and limitations of plasmonic-based approaches for MNP detection, providing real-world examples or studies that showcase their effectiveness or challenges.
-
Provide recent examples of significant advancements in SERS-based techniques for MNP detection, further emphasizing its potential as the most promising approach.
-
Discuss the current limitations or barriers that hinder wider adoption of plasmonic and SERS-based methods, encouraging researchers to address these issues in their future studies.
-
Consider including a section on potential future directions and emerging technologies that could have a substantial impact on MNP detection and quantification.
- The importance of biopolymers can be discussed(Ref.https://doi.org/10.1016/j.surfin.2022.102349)
To enhance the article's overall impact, I suggest the following minor revisions:
-
Expand on the advantages and limitations of plasmonic-based approaches for MNP detection, providing real-world examples or studies that showcase their effectiveness or challenges.
-
Provide recent examples of significant advancements in SERS-based techniques for MNP detection, further emphasizing its potential as the most promising approach.
-
Discuss the current limitations or barriers that hinder wider adoption of plasmonic and SERS-based methods, encouraging researchers to address these issues in their future studies.
-
Consider including a section on potential future directions and emerging technologies that could have a substantial impact on MNP detection and quantification.
- The importance of biopolymers can be discussed(Ref.https://doi.org/10.1016/j.surfin.2022.102349)
Author Response
Expand on the advantages and limitations of plasmonic-based approaches for MNP detection, providing real-world examples or studies that showcase their effectiveness or challenges.
A new paraprapgh was added at the end of section 3 - Plasmonic nanomaterials for MNPs detection. Paragraph 3.4 discusses the main advantages and disadvantages of the plasmonic techniques. It has to be stressed that only a small number of papers was published so far on this topic, as we mentioned in the previous paragraph 3.1, 3.2 and 3.3. We thus highlighted that this field is promising, but still in a really early stage of development.
Provide recent examples of significant advancements in SERS-based techniques for MNP detection, further emphasizing its potential as the most promising approach.
Almost all the cited papers in section 4 were published in the last 3 years. We added some sentences in the conclusion section to stress this fact. The field of plasmonic and SERS detection of MNP is still at its dawning, and we believe that more studies will be published soon. In this review we cited more than 75 papers published in 2020 or later: this is in our opinion and evidence of a quickly evolving field. Sentences stressing the novelty of the field were added also in paragraphs 4.2.4 and 4.3, not present in the original submission of the manuscript.
Discuss the current limitations or barriers that hinder wider adoption of plasmonic and SERS-based methods, encouraging researchers to address these issues in their future studies.
Limitations on the use of plasmonic methods were discussed in paragraph 3.4, while limitations of SERS methods are mentioned in paragraph 4.3. As mentioned before, plasmonic methods are in a very early stage of development and still needs a deeper analysis of advantages and limitations. SERS methods, on the other hands, showed many promising results, but only a few papers tested real samples and complex matrices.
Consider including a section on potential future directions and emerging technologies that could have a substantial impact on MNP detection and quantification.
A note on future directions was added at the end of paragraph 4.3, added in this revised version of the paper. It is not easy to make a comparison between the different techniques due to the lack of standard protocols and legislation in this field. A complete study of the different techniques is out of the scope of this review, but we wanted to give a summary of the most common approaches in table 1, providing all the relevant references to the reader.
The importance of biopolymers can be discussed(Ref.https://doi.org/10.1016/j.surfin.2022.102349)
A sentence was added in the introduction, mentioning the importance of biopolymers as substitutes of petroleum-derived plastics and as materials for MNP removal. We also added some references, including the paper suggested by the reviewer. We believe that, in spite of the clear importance of the topic, an in deep discussion of the use of biopolymers is out of the scope of this review.
Reviewer 2 Report
The authors presented a review article on micro- and nano-plastics detection using plasmonic nanomaterials and mainly focused on recent advancement leveraging SERS. The topic is important and urgent, as plastic pollution is becoming a threat to both human and wildlife. The article is generally organized, and detailed when describing primary research articles. I would recommend publishing the manuscript if the following comments can be addressed.
-
References should be added when the authors are explaining certain terminologies, such as localized surface plasmon resonance in Line 102-110, surface plasmon resonance in Line 140-148, and SERS effect in line 249-252.
-
In Line 185, the authors mentioned that enhancement factor was calculated in the cited article (Reference 46). The original article did show that ER immobilization in the Ni-NTA column enhanced the separation of PS from PVC and PE by affinity chromatography, but the usage of the term “enhancement factor” here can be confusing, because “enhancement factor” is defined as the enhancement of Raman signal by SEPS in the manuscript.
-
Summarized in Table 3, enhancement factor using the same substrate varies based on the colloidal size of the plastics. Can the author explain the reason? How does micro- or nano-plastics size, plasmon nanomaterial size and shape (eg. AuNPs vs Au nanorod) affect enhancement factor?
-
Most of the original research summarized in the manuscript detected plastics with well-defined size and materials within a simple matrix. It would be helpful if the authors can discuss the possible approaches to detect and quantify diverse plastic materials with unknown size and composition in complex sample matrices.
The manuscript is well written and only minor editing to correct typos is needed.
Author Response
References should be added when the authors are explaining certain terminologies, such as localized surface plasmon resonance in Line 102-110, surface plasmon resonance in Line 140-148, and SERS effect in line 249-252.
References explaining the terms were added in lines 104 (ref 31), 137 (ref 38) and line 250 (ref 111-113)
In Line 185, the authors mentioned that enhancement factor was calculated in the cited article (Reference 46). The original article did show that ER immobilization in the Ni-NTA column enhanced the separation of PS from PVC and PE by affinity chromatography, but the usage of the term “enhancement factor” here can be confusing, because “enhancement factor” is defined as the enhancement of Raman signal by SEPS in the manuscript.
We thank the referee for the observation. Reference to the EF was actually misleading and was removed from the text.
Summarized in Table 3, enhancement factor using the same substrate varies based on the colloidal size of the plastics. Can the author explain the reason? How does micro- or nano-plastics size, plasmon nanomaterial size and shape (eg. AuNPs vs Au nanorod) affect enhancement factor?
Some observations on the effect of different plasmonic nanomaterials were present in section 4.2. We added a new paragraph (4.2.4) with a deeper discussion on these effects and on the influence of micro and nanoplastic size on the enhancement factor. Two of the previously cited papers are discussed in detail: one examining the effect of different size plastics on the enhancement factor (SERS intensities are higher for lower size plastics) and the second discussing the effect of gold nanoparticles of two different sizes on the signal intensity (bigger particles increase the signal and thus decrease the detection limit).
Most of the original research summarized in the manuscript detected plastics with well-defined size and materials within a simple matrix. It would be helpful if the authors can discuss the possible approaches to detect and quantify diverse plastic materials with unknown size and composition in complex sample matrices.
As the reviewer correctly pointed out, most of the published research so far employs simple matrices, and only in a few cases real samples are studied. This is certainly related to the fact that research in this field is still at an early stage. This was pointed out in a new paragraph added to the review, paragraph 4.3.
Herein we tried to summarize and discuss the main limitations of the presented works and the future perspectives and challenges. A deeper study of real and complex samples and matrices is for sure one of the key points to be addressed.
Reviewer 3 Report
The manuscript entitled "Plasmonic nanomaterials for micro- and nanoplastics detection" represents a valuable work dealing with the use of plasmonic noble metal materials for the detection of micro- and nanoplastics. These plasmonic nanomaterials present some peculiar optical end electronic properties, which make them suitable for application using surface plasmon resonance, plasmon enhanced fluorescence, UV-Vis spectroscopy and surface enhanced Raman scattering (SERS) approaches. Their main advantages and drawbacks are also discussed in the manuscript.
All sections are well-organized and presented. Figures are very suggestive and representative for the topic under review. The references section is appropriate in relation with the addressed topic.
I have some minor comments to make:
- there are some minor editing errors; please, verify carefully the text and tables;
- the title for Table 1: this information is already given in the text preceding Table 1, so I think it can be removed from the title ("For each technique we give an indication of the size range of particles detected, we indicate if the technique can give qualitative information on the chemical composition and, if available, the limits of detection and/or quantification, based on literature references reported in the table. The main limitations of each technique are also reported in the table." );
- lines 265, 268: please, verify the equations to be correctly written;
- regarding EF formulations: I suggest to consider presentation of equations (2-5) as tabular data for an ease following and clarity for comparison purposes.
The manuscript can be considered for publication after resolving these minor aspects.
Author Response
- there are some minor editing errors; please, verify carefully the text and tables;
The text has been carefully edited, we hope to have fixed all the issues.
- the title for Table 1: this information is already given in the text preceding Table 1, so I think it can be removed from the title ("For each technique we give an indication of the size range of particles detected, we indicate if the technique can give qualitative information on the chemical composition and, if available, the limits of detection and/or quantification, based on literature references reported in the table. The main limitations of each technique are also reported in the table." );
We thank the reviewer for the observation, the table title was edited according to the suggestion.
- lines 265, 268: please, verify the equations to be correctly written;
We can’t understand how to modify the formula in line 265-268. The present formulas, joined to the explanatory text, seem to be adequate. Nevertheless we are ready to modify them if the referee indicates how to do it.
- regarding EF formulations: I suggest to consider presentation of equations (2-5) as tabular data for an ease following and clarity for comparison purposes.
Equations 2-5 were included in a table as suggested.
The manuscript can be considered for publication after resolving these minor aspects.
We thank the reviewer for the careful reading of this manuscript and for the editing suggestions.
Round 2
Reviewer 2 Report
All comments have been properly addressed.